# Comparison of the Immune Effects of an mRNA Vaccine and a Subunit Vaccine against Herpes Zoster Administered by Different Injection Methods

**DOI:** 10.3390/vaccines11051003

**Published:** 2023-05-20

**Authors:** Kangyang Lin, Han Cao, Ning Luan, Yunfei Wang, Jingping Hu, Cunbao Liu

**Affiliations:** Institute of Medical Biology, Chinese Academy of Medical Sciences and Peking Union Medical College, Kunming 650118, China; linky6679@163.com (K.L.); caohan@imbcams.com.cn (H.C.); luanning@imbcams.com.cn (N.L.); wangyf@imbcams.com.cn (Y.W.); hujingping@student.pumc.edu.cn (J.H.)

**Keywords:** herpes zoster, varicella-zoster virus, subcutaneous injection, intramuscular injection, mRNA vaccine, subunit vaccine

## Abstract

Previous studies have shown that the herpes zoster subunit vaccine Shingrix™ performs well in clinical trials. However, the key ingredient in its adjuvant, QS21, is extracted from rare plants in South America, so vaccine production is limited. Compared with subunit vaccines, mRNA vaccines have the advantages of faster production and not requiring adjuvants, but currently, there is no authorized mRNA vaccine for herpes zoster. Therefore, this study focused on herpes zoster subunit and mRNA vaccines. We prepared a herpes zoster mRNA vaccine and compared the effects of vaccine type, immunization route, and adjuvant use on vaccine immunological efficacy. The mRNA vaccine was injected directly into mice via subcutaneous or intramuscular injection. The subunit vaccine was mixed with adjuvants before immunization. The adjuvants include B2Q or alum. B2Q is BW006S + 2395S + QS21. BW006S and 2395S are phosphodiester CpG oligodeoxynucleotides (CpG ODNs). Then, we compared the cell-mediated immunity (CIM) and humoral immunity levels of the different groups of mice. The results showed that the immune responses of mice inoculated with the mRNA vaccine prepared in this study were not significantly different from those of mice inoculated with the protein subunit vaccine supplemented with the B2Q. The mRNA vaccine-induced immune responses following subcutaneous or intramuscular injection, and the different immunization routes did not lead to significant differences in immune response intensity. Similar results were also observed for the protein subunit vaccine adjuvanted with B2Q but not alum. The above results suggest that our experiment can provide a reference for the preparation of mRNA vaccines against herpes zoster and has certain reference significance for the selection of the immunization route; that is, there is no significant difference in the immune response caused by subcutaneous versus an intramuscular injection, so the injection route can be determined according to the actual situation of individuals.

## 1. Introduction

Varicella-zoster virus (VZV), also known as human herpes virus 3 (HHV-3), belongs to the alphaherpesviridae subfamily of herpesviridae, making it a double-stranded DNA virus [1]. Spread mainly through respiratory droplets and contact, VZV is highly contagious, especially among people with compromised immunity, such as children, the elderly and immunosuppressant users [2,3,4]. People are usually first infected with the virus in childhood, which causes chickenpox. After recovery, the chickenpox symptoms disappear, but the virus is not eliminated. Instead, it lurks in the trigeminal nerve and other dorsal root ganglia, where it can proliferate and cause herpes zoster in individuals with weakened immunity [5,6,7]. Antiviral drugs, such as acyclovir and interferon-alpha, are effective in the treatment of VZV infection [8]. However, herpes zoster can cause a variety of complications, such as herpes neuralgia, and the complications may last a long time, which seriously affects the quality of life of patients [9]. Therefore, it is necessary to develop a safe and effective vaccine that is easy to mass produce and to choose an appropriate route of inoculation to prevent the occurrence of herpes zoster.

There are two main vaccines available for shingles. One is Zostavax^®^ (Merck & Co., Inc., Kenilworth, NJ, USA), a live attenuated herpes zoster vaccine released in 2005, which is given as a single subcutaneous injection. The results of clinical trials of this vaccine showed that the protection rate was approximately 70% in people aged 50–59 years, but the effective rate dropped to less than 38% in people over 70 years old [10]. The other is Shingrix™ (Glaxo Smith Kline (GSK), Rockville, MD, USA), a subunit vaccine containing the VZV glycoprotein E (gE) protein, which is administered intramuscularly and became available in 2017. The results of clinical trials have shown that this vaccine has a protective efficiency of more than 90% in people aged 50–59 years or over 70 years and can reduce the risk of postherpetic neuralgia [11]. Shingrix™ provides significantly better protection than Zostavax^®^, especially in the 70+ age group. Shingles is caused by viruses lurking in the ganglia, which cannot be eliminated completely by VZV-specific antibodies produced by the body. Therefore, a specific cellular immune response plays a more important role in preventing shingles than does humoral immunity [3,7,12,13].

However, the immunogenicity of protein subunit vaccines is weak and may not be effective in stimulating specific cellular immune responses, so it is important to select appropriate adjuvants to ensure that subunit vaccines produce protective responses. The antigenic component in Shingrix^TM^ is the gE protein, which plays an important role in the course of VZV infection [14]. Shingrix^TM^ uses the AS01 adjuvant system, in which the key ingredient QS21 is derived from a rare plant that grows in South America and is therefore difficult to produce in large quantities [15]. This is the main reason why the vaccine is expensive and difficult to obtain. mRNA vaccines are a new type of vaccine. Taking the Pfizer-BioNTech BNT162b2 vaccine (Pfizer-BioNTech, Nierenzentrum Kronach, Germany) as an example, this vaccine can be synthesized in batches in vitro and has a self-adjuvanting property, which allows it to effectively induce cellular and humoral immunity without an adjuvant [16]. Therefore, we hope to solve a current problem in the large-scale production of herpes zoster subunit vaccines by using mRNA vaccines.

In addition to vaccine type, adjuvant type and immunization route are important factors affecting vaccine effectiveness. CpG ODNs, as an adjuvant, can be injected with vaccines to enhance the immune response and change the type of immune response [17]. In this study, B2Q (BW006S + 2395S + QS21, BW006S and 2395S are CpG ODNs) groups and alum groups were used to assess the subunit vaccine in the context of different adjuvants. The main reason for this is that we demonstrated in a previous study that BW006S combined with 2395S can make the vaccine-induced immune response more Th1 biased [18]. The cellular immune response plays an important role in eliminating infected cells and preventing herpes zoster. In terms of the immunization route, the large number of blood vessels in muscle tissue can provide a sufficient blood supply to allow the vaccine components to be transported throughout the body through blood circulation. The blood supply of subcutaneous tissue is not as rich as that of muscle tissue, but there is sufficient tissue fluid and lymphatic fluid to participate in the body fluid circulation. There are structural differences between the subcutaneous tissue and muscle tissue, which may lead to different immunological effects between vaccines injected into the two sites. However, whether and how the immune response to a shingles vaccine differs in mice immunized via the muscular versus subcutaneous route is unknown, which is the focus of this study.

## 2. Materials and Methods

### 2.1. Vaccine Preparation

Two types of vaccines were prepared in this study, namely, mRNA and subunit vaccines. It is known that mRNA is easily degraded and, as a negatively charged hydrophilic macromolecule, does not easily pass through the cell membrane. Lipid nanoparticles (LNPs) can be used to encapsulate mRNA to avoid nuclease hydrolysis, and the positively charged lipid inclusion formed is conducive to mRNA passing through the negatively charged cell membrane and entering the cell to exert activity [19]. Raw materials for the preparation of the mRNA vaccine were provided by Nanjing Vazyme Biotech Co., Ltd. A total of 360 μg of gE mRNA was required based on 15 μg of gE mRNA per dose of mRNA vaccine. According to previous experiments, the rate of gE mRNA encapsulation by LNPs is usually over 90%, so a total of 400 μg of gE mRNA prepared in this experiment was sufficient to meet the experimental requirements. The mRNA was obtained through in vitro amplification, and impurities in the product mRNA were removed by magnetic bead purification. The purified mRNA concentration was measured, and the degree of mRNA degradation was detected by electrophoresis. The aqueous phase of the mRNA vaccine was prepared using citrate buffer, followed by preparation of the lipid phase, which was one-third the volume of the aqueous phase (the constituents of the lipid phase included MC3, DSPC, cholesterol and DMG-PEG2000, with a molar ratio of 50:10:37.5:2.5). Using a microfluidic apparatus, the aqueous phase containing mRNA was coated by the lipid phase to form LNPs, and the inclusions were placed in an ultrafiltration tube containing a PBS solution (treated with DEPC) for ultrafiltration. In conclusion, the LNPs vaccine-blocking process was similar to that of previous studies [18,20]. The size and polydispersity index (PDI) of the mRNA vaccine were measured by Malvern ZEN3600 (Malvern Instruments Ltd., Worcestershire, UK). After the prepared mRNA vaccine was split overnight, the amount of enveloped mRNA was quantitatively detected with the Quant-iT^TM^ RiboGreen^TM^ RNA Assay Kit (Thermo Fisher, Eugene, OR, USA) to calculate the envelopment rate. The rest of the vaccine was stored at −80 °C to avoid mRNA degradation.

The subunit vaccine used in this study consisted of antigenic proteins and adjuvants that were mixed before immunization. We selected a gE extracellular region protein expressed by Chinese hamster ovary (CHO) cells as the protein antigen (AtaGenix Laboratory Co., Ltd., Wuhan, China). For the adjuvant, in addition to the commonly used alum adjuvant, a CpG ODNs adjuvant was selected. The CpG ODNs used in this study included BW006S (5′-tcg acg ttc gtc gtt cgt cgt tc-3′) and 2395S (5′-tcg tcg ttt tcg gcg cgc gcc g-3′) were synthesized by Sangon Biotech (Shanghai, CHN). Because QS21 is an immunostimulatory saponin and a key adjuvant component of Shingrix™, it was administered with the two CpG ODNs described above. QS21 was supplied by Alpha Diagnostic Intl. Inc. (San Antonio, TX, USA). The vaccine components were formulated according to the information in Table 1 before administration.

### 2.2. Mouse Studies

Specific pathogen-free C57BL/6 mice were provided by the Department of Small Animal Laboratory, Institute of Medical Biology, Chinese Academy of Medical Sciences. All mice were female mice aged 6 to 8 weeks and weighing 16 to 18 g. A total of 42 mice were randomly divided into 7 groups (N = 6): 2 groups of mRNA vaccine-immunized mice, 4 groups of subunit vaccine-immunized mice and 1 group of PBS-immunized mice as a negative control (Table 1). The animals were divided into groups and immunized: The mice in groups 1, 3 and 5 were injected intramuscularly near the tibia, and the mice in groups 2, 4 and 6 were injected subcutaneously in the back of the neck. Vaccines were administered every 4 weeks for a total of 2 injections. Two weeks after the final immunization, the mice were anesthetized, the chest and abdomen were opened, blood was collected by cardiac puncture and centrifuged at 3500 rpm for 20 min, and the serum was separated. Mouse spleens were removed and treated as described previously to obtain a single-cell suspension of spleen cells [21].

### 2.3. gE-Specific IgG Titer Detection

The level of specific antibodies against the VZV gE protein in mouse serum was determined by enzyme-linked immunosorbent assay (ELISA). The gE protein was diluted to 2 μg/mL with PBS, thoroughly mixed, and added to 96-well plates (Corning Inc., Corning, NY, USA) at 50 μL/well, which were incubated overnight in a refrigerator at 4 °C. The liquid in the plates was discarded, and the plates were washed with PBST. Five percent (*w*/*v*) skim milk powder was used as the blocking solution and incubated at 37 °C for 1 h. The blocking solution was discarded, and the plates were washed with PBST. Gradient diluted mouse serum was added to the wells and incubated at room temperature for 2 h. A goat anti-mouse IgG–horseradish peroxidase conjugate (GM-HRP; 1:10,000, Bio-Rad, Hercules, CA, USA) was added as the secondary antibody to detect the binding antibody and incubated at 37 °C for 1 h. Then, the plates were washed with PBST, and the substrate 3,3′,5,5′-tetramethylbenzidine (TMB; BD, San Diego, CA, USA) was added. After 5 min, 2 mol/L sulfuric acid was added to stop the reaction. Finally, the absorbance at 450 nm was measured by a spectrophotometer (BioTek Instruments, Inc., Winooski, VT, USA). Antibody titers were defined as the maximum dilution of serum with an absorbance greater than 0.15 at 450 nm. For the convenience of calculation, the antibody titer of samples with absorbance less than 0.15 at 450 nm at a dilution ratio of 1:2000 was uniformly recorded as 100.

### 2.4. Cytokine Analysis

The levels of the cytokines IL-2 and IFN-γ secreted by mouse spleen cells were detected by ELISA with the double-antibody sandwich method. The capture antibody was diluted with PBS to the appropriate concentration (3 μg/mL for IL-2 and 4 μg/mL for IFN-γ). Prepared spleen cells were added to 96-well plates (1 × 10^6^ cells per well), and gE protein was added to the wells other than the negative and positive control wells as a stimulus. PMA + ionomycin (10 μg/mL, DAKEWE, Beijing, China) was added to the 96-well plates (Corning Inc., Corning, NY, USA) as a positive control. The cells were incubated overnight at 37 °C and 5% CO_2_, and the cell supernatants were collected to detect the levels of IL-2 and IFN-γ [22].

### 2.5. Enzyme-Linked Immunospot (ELISPOT) Assay

Test plates were activated with ethanol and then washed to remove any residual ethanol. Specific antibodies against IL-2 and IFN-γ were added to the plates at a concentration of 2 μg/mL, and the plates were refrigerated at 4 °C overnight. The liquid in the wells was removed, the plates were washed with 1640 complete medium, and then fresh 1640 complete medium was added prior to incubation for 2 h. After the blocking solution was discarded, 30 μL of spleen cell suspension (containing 3 × 10^5^ spleen cells), 70 μL of 1640 complete medium, and 50 μL of gE antigen (at a concentration of 60 μg/mL) were added to each well. In the negative control well, 30 µL of PBS group mouse cells and 120 µL of medium were added. In the positive control well, PMA + ionomycin was used as a stimulus. The 96-well plates were cultured overnight at 37 °C and 5% CO_2_ without movement. The plates were then centrifuged at 800× *g* for 5 min, and the liquid in the plates was discarded. Next, cold deionized water was added, and the plates were allowed to soak. Then, 1 µg/mL of the secondary antibody was added in 50 µL per well, and the plates were incubated at room temperature for 2 h in the dark. After plate washing, diluted HRP-conjugated Streptavidin (1:1500) was added and incubated for 1 h at room temperature. After washing, a substrate solution was added and incubated at room temperature in the dark for 15–30 min. Then, the plates were rinsed with running water to stop color development and stored upside down overnight to dry before being read (Autoimmun Diagnostika GmbH, Strassberg, Germany). The experiments were performed using an ELISPOT assay kit (BD).

### 2.6. Flow Cytometry

Spleen cells (1 × 10^6^) and gE protein (10 µg/mL) were added to each well and incubated at 37 °C and 5% CO_2_ for 2 h. Brefeldin A was added to the wells at a final concentration of 5 µg/mL and incubated overnight. The cells were washed the next day with buffer, and 100 μL of Zombie NIRTM dissolved in DMSO was added to each well. After washing with PBS, 50 μL of 5 μg/mL anti-CD16/CD32 antibodies were added to each well and incubated at 4 °C for 10 min to block nonspecific binding to Fc receptors. PerCP-Cy5.5-conjugated anti-mouse CD4 was added to the samples successively and incubated at 4 °C for 30 min for surface maker staining. Then, washing was performed, and fixation buffer was added to the samples and incubated at room temperature for 20 min to fix the cell membrane. Intracellular antibodies (PE-conjugated anti-mouse IFN-γ and APC-conjugated anti-mouse IL-2 antibodies) diluted in permeabilization wash buffer were added for intracellular staining after washing. For staining, the samples were incubated at room temperature in the dark for 40 to 60 min. After staining, a CytoFLEX flow cytometer (Beckman, Indianapolis, IN, USA) was used to analyze CD4^+^ T cells in the samples. The reagents used in the experiment were all obtained from BioLegend (San Diego, CA, USA).

### 2.7. Statistical Analysis

All experimental data in this paper were analyzed with unpaired *t*-tests using GraphPad Prism 9.4.0 software (* *p* ≤ 0.05; ** *p* ≤ 0.01; *** *p* ≤ 0.001; **** *p* ≤ 0.0001; ns, no significant difference, *p* > 0.05).

## 3. Results

### 3.1. Almost All gE mRNA Was Coated by LNPs to Form Particles with a Uniform Diameter

The lipid phase encapsulated mRNA into LNPs with a microspherical structure and its pattern is shown in Figure 1A. The particle size, encapsulation rate and polydispersity index (PDI) of the mRNA vaccine prepared in this study were consistent with our previous experiments [18,20]. The mean diameter of the LNPs vaccine was approximately 80 nm (Figure 1B), and the PDI was 0.116 (Figure 1C). A small amount of vaccine was cleaved overnight to detect the mRNA concentration. By calculating the ratio between the quality of detected mRNA and the input dose, the encapsulation efficiency of the vaccine was obtained, and the result was 102.8% (Figure 1D). The encapsulation rate was slightly higher than 100%, which may be due to experimental deviation. If calculated as 100%, the actual content of gE mRNA was about 16.7 μg per injection. Agarose gel electrophoresis showed that the mRNA in the LNPs vaccine degraded only slightly after overnight cleavage, indicating good integrity (Figure 1E).

### 3.2. The Effect of the Injection Route on Humoral Immune Response Intensity Was Related to Vaccine Type and Adjuvant Composition

The IgG antibody level is a commonly used indicator to evaluate the humoral immune response. Therefore, the VZV gE protein-specific IgG antibody titer was used to determine whether different groups of mice had different humoral immune responses due to differences in the immunization route, vaccine type and adjuvant (Figure 2). VZV gE protein-specific antibodies were detected in the serum of all mice in the six experimental groups but not in that of mice in the PBS group. In the mRNA vaccine groups, the mean serum titers of VZV gE protein-specific antibodies were 512,000 and 725,333 in the intramuscular and subcutaneous injection groups, respectively, and 640,000 and 597,333 in the protein subunit vaccine groups supplemented with the adjuvant B2Q. Statistical analysis results generated with an unpaired *t*-test found no significant difference between the group mRNA (I.M.) and group mRNA (S.C.), and the group Subunit + B2Q (I.M.) and group Subunit + B2Q (S.C.) were also not significantly different, suggesting that for the mRNA vaccine and the subunit vaccine supplemented with the adjuvant B2Q, the strength of the humoral immune responses induced by subcutaneous and intramuscular injections was comparable. In other words, the route of injection may not be the key factor affecting the humoral immune response strength. Interestingly, the levels in group 5 (Subunit + Alum (S.C.)) were significantly higher than that in group 6 (Subunit + Alum (I.M.)) (* *p* ≤ 0.05), suggesting that for the subunit vaccine used in this study, when the adjuvant was changed from B2Q to alum, the influence of the injection route on the humoral immune response increased.

### 3.3. Cellular Immune Responses Induced by Intramuscular or Subcutaneous Injections of the mRNA or Subunit Vaccines Were Not Significantly Different

The levels of IL-2 and IFN-γ secreted by spleen cells were detected by ELISA. The PBS group was the negative control, and the other six groups were the experimental groups. The IL-2 levels in group 1 (4.682 µg/mL), group 2 (8.472 µg/mL), group 3 (6.360 µg/mL) and group 4 (5.554 µg/mL) were significantly higher than those in the negative control group (0.1228) ug/mL), group 5 (0.1348 µg/mL) and group 6 (0.1704 µg/mL). Similarly, the IFN-γ levels of group 1 (40.53 µg/mL), group 2 (56.11 µg/mL), group 3 (65.09 µg/mL) and group 4 (57.91 µg/mL) were much higher than those of the negative control group (0.5034 µg/mL), group 5 (0.6030) ug/mL) and group 6 (1.191 µg/mL). This suggests that mRNA vaccines and subunit vaccines supplemented with adjuvant B2Q induce strong cellular immune responses, while alum-adjuvanted subunit vaccines do not.

Furthermore, the unpaired *t*-test results showed that there was no significant difference in IL-2 levels between group 1 and group 2 (*p* = 0.1105) (Figure 3A) and no significant difference in IFN-γ levels (*p* = 0.0558) (Figure 3B). There was no significant difference in IL-2 levels between group 3 and group 4 (*p* = 0.6435) (Figure 3A) and no significant difference in IFN-γ levels (*p* = 0.5725) (Figure 3B).

In the detection of cytokine levels, ELISPOT, as a reliable method, can be used for mutual verification with ELISA, making an experimental conclusion more credible. ELISPOT results showed that the number of spleen cells producing IL-2 in group 1 (620.8), group 2 (623.0), group 3 (511.7) and group 4 (482.0) was significantly higher than that in the negative control group (175.0). Group 5 (210.0, ns) and group 6 (172.2, ns) were basically consistent with the negative control group. However, administration of the same vaccine via different routes did not lead to a significant difference in the number of spleen cells that produced IL-2 (Figure 3C). In Figure 3E, representative pictures are displayed. Similarly, the number of spleen cells producing IFN-γ in group 1 (726.0, *** *p* ≤ 0.001), group 2 (901.0, **** *p* ≤ 0.0001), group 3 (1067, **** *p* ≤ 0.0001) and group 4 (918.3, **** *p* ≤ 0.0001) was significantly different from that in the negative control group (138.3) but not that in group 5 (121.2, ns) or group 6 (113.3, ns) (Figure 3D). In Figure 3E, representative pictures are displayed. In short, the ELISPOT results were consistent with those of the ELISA analyses, both of which indicated that the mRNA vaccine and the subunit vaccine supplemented with the adjuvant B2Q could effectively induce cellular immune responses, while the subunit vaccine adjuvanted with alum could not. Importantly, this ability did not change significantly depending on whether the injections were intramuscular or subcutaneous.

This phenomenon was also observed by flow cytometry (Figure 4). The frequencies of CD4^+^ T cells producing IL-2 and IFN-γ were not increased in the alum adjuvant group compared with the negative control group. For subunit vaccines supplemented with the adjuvant B2Q, there was no significant difference in the percentage of CD4^+^ T cells producing cytokines (IL-2 and IFN-γ) between subcutaneously and intramuscularly injected mice. Although intramuscular injection resulted in a higher percentage of IL-2-producing CD4^+^ T cells than subcutaneous injection for the mRNA vaccines, the difference may not exist after discarding the outlier.

### 3.4. Different Types of Vaccines with the Same Injection Methods Can Induce an Immune Response of Comparable Intensity

An unpaired *t*-test was used to compare the ELISA, ELISPOT and flow cytometry results of mRNA vaccine and subunit vaccine supplemented with B2Q. Antibody titers of group 1 and group 3 were 512,000 and 640,000, *p* = 0.4732, and those of group 2 and group 4 were 725,333 and 597,333, *p* = 0.4506 (Figure 2H). Cytokine analysis showed that IL-2 levels of group 1 and group 3 were 4.682 µg/mL and 6.360 µg/mL, *p* = 0.3663, and those of group 2 and group 4 were 8.472 µg/mL and 5.554 µg/mL, *p* = 0.1939 (Figure 3A). The IFN-γ levels of group 1 and group 3 were 40.53 µg/mL and 65.09 µg/mL, *p* = 0.0146, and those of group 2 and group 4 were 56.11 µg/mL and 57.91 µg/mL, *p* = 0.8792 (Figure 3B). The ELISPOT results showed that the number of spleen cells producing IL-2 in group 1 and group 3 was 620.8 and 511.7, *p* = 0.1074, and that in group 2 and group 4 was 623.0 and 482.0, *p* = 0.2574 (Figure 3C). The number of spleen cells producing IFN-γ in group 1 and group 3 was 726.0 and 1067 (*p* = 0.0248), and 901.0 and 918.3 (*p* = 0.9201) in group 2 and group 4 (Figure 3D). Flow cytometry showed that the percentage of CD4^+^ T cells producing IL-2 in group 1 and group 3 was 2.353% and 1.422%, *p* = 0.2970, and that in group 2 and group 4 was 1.582% and 1.015%, *p* = 0.0537 (Figure 4B). The percentages of CD4^+^ T cells producing IFN-γ in group 1 and group 3 were 3.120% and 1.967% (*p* = 0.4137), and 1.833% and 1.587% (*p* = 0.5436) in group 2 and group 4 (Figure 4C). The results of the above different experiments have a good consistency. Comprehensive analysis of all data is not difficult to find that there is no significant difference in humoral immunity and cellular immunity induced by mRNA vaccine and subunit vaccine adjuvanted with B2Q when the same injection method is used.

## 4. Discussion

In this study, herpes zoster protein subunit vaccines and mRNA vaccines prepared by us were injected into mice via subcutaneous and intramuscular routes. After completion of the immunization process, serum antibody titer detection, cytokine detection, ELISPOT analysis, flow cytometry analysis and other experiments were carried out. The data suggested that the influence of the immunization route on the vaccine effect was not invariable; instead, it was complex and needs to be discussed according to specific circumstances [23]. The immunization route was not the key factor affecting the vaccine effect in both the protein subunit vaccine group with BW006S + 2395S + QS21(B2Q) adjuvant and the mRNA vaccine group, in which the antibody levels, cytokine levels and other indicators of mice did not show significant differences with the change in injection method.

In contrast, though very weak cellular immunities were detected through both injection methods, the humoral immune response induced by subcutaneous injection of the protein subunit vaccine with the alum adjuvant was stronger than that induced by intramuscular injection.

While mRNA vaccines do not require adjuvants to effectively induce humoral and cellular immune responses, subunit vaccines are usually administered with adjuvants. Therefore, the selection of adjuvant has an important impact on the protective effect of the subunit vaccine, which is also confirmed in this study. When B2Q was used, the subunit vaccine effectively induced cellular and humoral immune responses, and its intensity was similar to that of the mRNA vaccine group. We speculated that the mechanism of action of the subunit vaccine supplemented with B2Q was similar to that of the mRNA vaccine to some extent. QS21 is ingested by cells through cholesterol-dependent endocytosis, which can damage the plasma membrane and release the contents from the endocytosomes into the cytoplasm, resulting in the cross-presentation of antigens. At the same time, the CpG ODNs component in B2Q may act as a ligand of toll-like receptor (TLR) to stimulate nucleic acid receptors in the cytoplasm, e.g., activate the cGAS-sting signaling pathway, and induce IFN-dependent antiviral immunity. The role of CpG ODNs in cellular immunity lies in their ability to induce the production of cytokines such as IL-12 and IFN-γ, which promote Th1 cell differentiation [24,25].

At present, mRNA vaccines are mainly administered by intramuscular injection, but in this experiment, it can be seen that the immune effect induced by subcutaneous injection is not lower than intramuscular. So, subcutaneous injection may be an alternative route for mRNA vaccines because it has been shown that subcutaneous injection is associated with fewer systemic adverse events (AEs) than intramuscular injection [26].

Common vaccination methods include not only intramuscular injection and subcutaneous injection, as mentioned in this article, but also intradermal injection and nasal mucosal immunization. Although intradermal immunization is regarded as a potential vaccination route, it has been difficult to widely use this method due to the difficulty of drug administration, which creates high demands on operators. In addition, studies have shown that compared with subcutaneous injection, intradermal injection of the same dose of VZV vaccine can cause a stronger humoral immune response, but whether the cellular immune response improves requires further evidence, and intradermal injection has a higher incidence of erythema and other adverse reactions [27]. Mucosal immunization is usually used with live attenuated vaccines that require stimulation of an effective mucosal immune response, but an effective cellular immune response is key in the prevention of shingles. Therefore, no experiments on intradermal injection or mucosal immunization were designed in this study. To be honest, we selected only the two most common injection routes and did not evaluate the effects of some emerging vaccination routes, such as microarray patches (MAPs) [28], which may lead to some limitations in this study. In addition, although this study identified the influence of the immunization route on the effectiveness of the two different forms of herpes zoster vaccines, the mechanism underlying this influence was not explored, and thus it may need further research in our future studies.

## 5. Conclusions

In summary, we prepared a candidate protein subunit vaccine against shingles that produced an immune response comparable to that achieved with an mRNA candidate vaccine, whether administered subcutaneously or intramuscularly. In addition, we acknowledge that the immunization route may influence vaccine effectiveness, but does not appear to exert an effect all the time, at least not as a key factor in the immune effectiveness of the two vaccines used in this study. For the VZV-gE protein subunit vaccine, the choice of vaccine adjuvant may be the decisive factor in the effectiveness of shingle prevention. Currently, there is no mRNA vaccine against herpes zoster on the market. Our experimental results provide a reference for future research: subcutaneous and intramuscular injection routes are both optional immunization methods, which can be selected according to the actual situation without affecting the immune effect of the vaccine.

## Figures and Tables

**Figure 1 vaccines-11-01003-f001:**
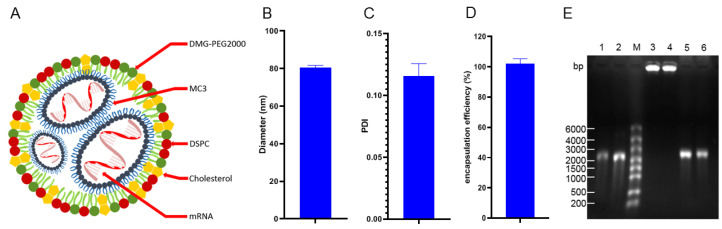
Model diagram and characteristics of the lipid nanoparticle (LNPs) vaccine. (**A**) Schematic of the LNPs vaccine. (**B**) Diameter of the LNPs vaccine. (**C**) Polydispersity index (PDI, an indicator of the homogeneity of response particle size) of the LNPs vaccine. (**D**) Encapsulation efficiency of the LNPs vaccine. (**E**) Electrophoresis results to assess whether the preparation process for the LNPs vaccine resulted in mRNA degradation; 1 and 2: mRNA not encapsulated into the LNPs vaccine; M: marker with a molecular weight of 6000 bp; 3 and 4: mRNA encapsulated to form the LNPs vaccine; 5 and 6: mRNA released by cleavage of the LNPs vaccine.

**Figure 2 vaccines-11-01003-f002:**
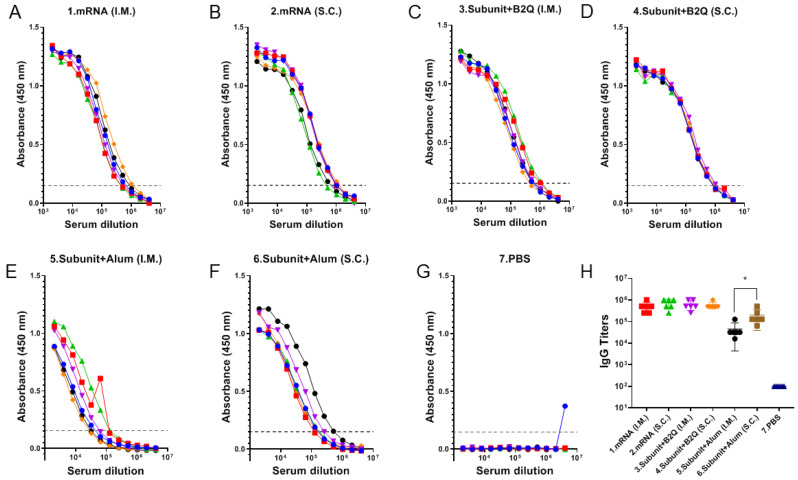
Detection of humoral immune responses in mice by enzyme-linked immunosorbent assay (ELISA). (**A**–**G**) The serum of each mouse in each group was serially diluted for detection. Each point represents the absorbance value at a wavelength of 450 nm at the corresponding dilution, each curve represents the data for one mouse, and the dashed line (y = 0.15) represents the cutoff signal for the IgG titer value. (**H**) IgG-specific antibody titer. Absorbance values at a wavelength of 450 nm were greater than the maximum dilution of 0.15, and each point represents data from one mouse. Statistical plots were drawn with GraphPad Prism 9.4.0, and the data displayed as the mean and standard deviation (SD) were analyzed by an unpaired *t*-test; * *p* ≤ 0.05.

**Figure 3 vaccines-11-01003-f003:**
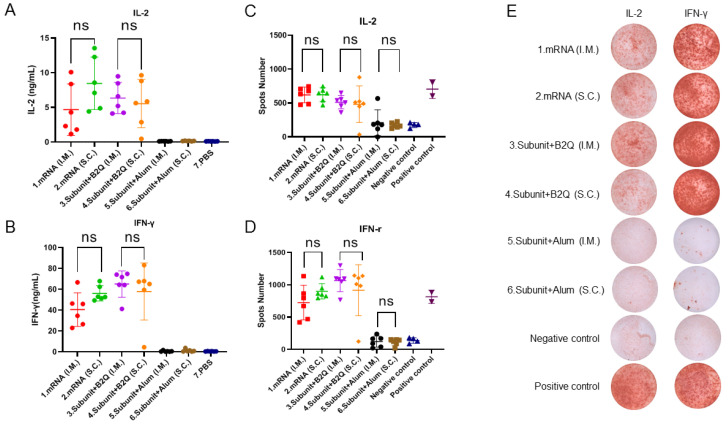
Cellular immune response levels in mice were measured by ELISA and enzyme-linked immunosorbent assay (ELISPOT). (**A**,**B**) ELISA detection of the concentrations of IL-2 (**A**) IFN-gamma (**B**) in the supernatants of mouse spleen cells treated with 10 µg/mL gE protein. (**C**,**D**) ELISPOT was used to determine the number of mouse splenocytes that produced IL-2 (**C**) and IFN-γ (**D**) after stimulation with 10 μg/mL gE protein. (**E**) Representative pictures of ELISPOT. Statistical plots were drawn with GraphPad Prism 9.4.0, and data are shown as the mean and standard deviation (SD). Unpaired *t*-test; ns, no significant difference, *p* > 0.05.

**Figure 4 vaccines-11-01003-f004:**
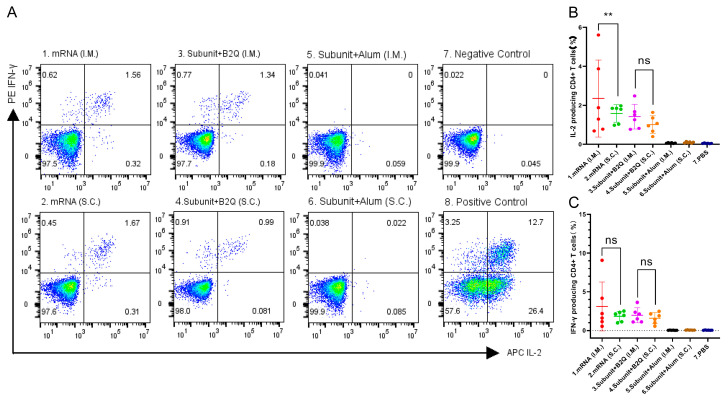
Flow cytometry results. (**A**) Representative pictures of flow cytometry. (**B**,**C**) The proportion of CD4^+^ T cells producing IL-2 (**B**) or IFN-γ (**C**) in mouse spleen cells stimulated with 10 µg/mL gE protein. Statistical plots were drawn with GraphPad Prism 9.4.0, and data are shown as the mean and standard deviation (SD). Unpaired *t*-test; ** *p* ≤ 0.01; ns, no significant difference, *p* > 0.05.

**Table 1 vaccines-11-01003-t001:** The grouping of animals and the composition of vaccines.

Vaccine Group	gE mRNA (μg)	gE Protein (μg)	CpG ODNs (μg)		QS21 (μg)	Alum	Injection
			BW006S	2395S			
1. mRNA (I.M.)	15	-	-	-	-	-	I.M.
2. mRNA (S.C.)	15	-	-	-	-	-	S.C.
3. Subunit + B2Q (I.M.)	-	10	5	5	5	-	I.M.
4. Subunit + B2Q (S.C.)	-	10	5	5	5	-	S.C.
5. Subunit + Alum (I.M.)	-	10	-	-	-	√	I.M.
6. Subunit + Alum (S.C.)	-	10	-	-	-	√	S.C.
7. PBS (I.M.)	-	-	-	-	-	-	I.M.

- Not added; √ Added; I.M., Intramuscular injection; S.C., Subcutaneous injection.

## Data Availability

All the data from the study are available from the corresponding author upon reasonable request.

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
