# Peer review of "Comparison of the Immune Effects of an mRNA Vaccine and a Subunit Vaccine against Herpes Zoster Administered by Different Injection Methods"

_vaccines, 2023, doi:10.3390/vaccines11051003_

Round 1

Reviewer 1 Report

The authors describe their experience using an mRNA vaccine for Herpes zoster administered via IM and SC routes with tandem observations of a subunit vaccine administered via the same routes with or without various adjuvant formulations.  

I found the experimental methods well designed and the inclusion of various test groups adds to the findings.  My main concern is that the authors focus nearly entirely in their analysis on the outcomes relative to the IM vs SC routes of administration and say little to nothing about the data relative to comparability of the subunit vaccine to mRNA vaccine only when using Th1 promoting adjuvants.  

This aspect of the study findings are highly intriguing and important and yet in the manuscript are given only cursory discussion.  The authors note the following:

1. "This phenomenon was also observed by flow cytometry (Figure 4). The frequencies of CD4+ T cells producing IL-2 and IFN-γ were not increased in the alum adjuvant group compared with the negative control group."

2.  "This suggests that mRNA vaccines and subunit vaccines supplemented with the adjuvant B2Q induce strong cellular immune responses, while alum-adjuvanted subunit vaccines do not."

Little to no follow up of these observations or discussion takes place in the discussion section of the manuscript, which is focused primarily on the comparison and contrast of results based on injection method alone.

I believe that a more extensive discussion of these findings is warranted and that the authors have completed studies with an adequate design that would allow for this to be undertaken.  

For example, can the authors address the following:

1.  Implications of use of CpGODN containing adjuvant in promoting different immune system, particularly cellular activation.

2.  Discussion of finding relative to comparison of alum and CpGODN and implications of immunization efficacy in the short and long term.

I believe that these elements will strongly boost the papers interest to readers of the journal.

Reviewer 2 Report

1.     The manuscript # vaccines-2383223 compared immunization routes for mRNA, and adjuvants for protein subunit vaccines for herpes zoster in C57BL/6 mice. The vaccination was done either subcutaneous or intramuscular. Adjuvants for protein subunit vaccine were CpG, QS21 or alum. Both cellular and humoral immunity were assessed. Authors reported that the immune responses in mice inoculated with the mRNA vaccine were not significantly different from protein subunit vaccine supplemented with the QS21+ CpG ODN adjuvant. The mRNA vaccine following subcutaneous or intramuscular injection did not lead to significant differences in immune response intensity. Similar results were also observed for the protein subunit vaccine adjuvanted with QS21+CpG ODN but not alum.

2.     Abstract line 17, the study is introduced as CPG and alum adjuvants were used, but later line 27 indicates that QS21 is also used with CPG alum.  Adjuvant QS12 should be added on line 17.

3.     In the introduction (line 78) of this paper, additional adjuvants BW006S+2395S + QS21(B2Q) are introduced, without any description. It became clear in the method section that BW006S and 2395S are CpG ODNS. It should be made clear either in the abstract or in the introduction. 

4.     All immunogenicity studies were done in female mice only.  Sex as a biological variation was not considered.

5.     Overall, the manuscript is simply an immunogenicity study with mRNA or protein-adjuvanted based vaccine in C57BL/6 mice.

It is accepted.
